# Temporomandibular Joint and Otitis Media: A Narrative Review of Implications in Etiopathogenesis and Treatment

**DOI:** 10.3390/medicina58121806

**Published:** 2022-12-08

**Authors:** Edoardo Bernkopf, Giovanni Cristalli, Giovanni Carlo de Vincentiis, Giulia Bernkopf, Vincenzo Capriotti

**Affiliations:** 1Dental Clinic, Via Massaciuccoli 19, 00100 Rome, Italy; 2Otolaryngology Unit, Bambino Gesù Children’s Hospital, IRCCS, Via della Torre di Palidoro, 00050 Rome, Italy; 3Otorhinolaryngology and Head and Neck Surgery Unit, ASST Bergamo Ovest, Treviglio-Caravaggio Hospital, Piazzale Ospedale Luigi Meneguzzo 1, 20047 Treviglio, Italy

**Keywords:** otitis media, otitis media with effusion, recurrent acute otitis media, temporomandibular joint, temporomandibular joint dysfunction, oral appliance, gnathology, malocclusion, OSAS

## Abstract

Otitis media (OM) and its recurring (rAOM), effusive (OME), and chronic forms, represent a frequent clinical challenge. The middle ear, the mandible, and the temporomandibular joint (TMJ) share several embryological and anatomical connections. Despite that, the role of mandibular malposition and TMJ dysfunction is frequently overlooked in the management of otitis media. In this narrative review, we present current evidence supporting the etiopathogenetic role of a dysfunctional stomatognathic system in the onset of OM and the effectiveness of orthognathic treatment in preventing rAOM and OME. In particular, a focus on the influence of TMJ on Eustachian tube function is provided.

## 1. Introduction

Otitis media (OM) consists of inflammation in the middle ear. Subcategories include acute otitis media (AOM), otitis media with effusion (OME), recurrent acute otitis media (rAOM), and chronic suppurative otitis media (CSOM) [1]. While the first three entities are common conditions in childhood, the latter is more frequent among adults and is often the result of previous AOM or rAOM [2,3,4].

The etiopathogenesis is still debated, though it is thought to be multifactorial, with a central role attributed to the Eustachian tube (ET) function [5]. Other risk factors for OM are environmental (cigarette smoke, exposure to the community) and genetic factors, which, together with adenoid hypertrophy, allergy, upper respiratory tract infections, and reflux, could affect ET function [6]. Despite the thorough research in this field, these common conditions still pose a therapeutic challenge, often demanding repeated antimicrobial prescriptions with growing bacterial resistance issues and chronicization that could require surgical approaches with uncertain results [7].

The pathogenetic role of the temporomandibular joint (TMJ) function, mandible position, and malocclusion has been hypothesized in this complex scenario [8]. Relation between various ear conditions and TMJ has been known for a century, with a possible role for orthognathic therapy in ameliorating ear function [9,10,11]. Dental malocclusion has a high prevalence in children, ranging from 20 to 40%, with variations among different ethnic groups [8]. In an Italian epidemiological study on 800 children, Sfondrini, Bianchi, and Bricca found a very high prevalence of malocclusion (80.3%), with a prevalence of males (83%) over females (77%) [12]. Treatment is mainly orthodontic, with mandibular repositioning devices as the usual first step [8]. Focusing on OM, hints for a link between TMJ and dental problems come from the interdependent embryological and phylogenetic development, anatomical and physiologic connections, the high prevalence of OM in children with maxillo-facial malformations and reported amelioration of otologic disease when treating jaw malposition or occlusal problems [8,13]. The relationship between OM and the stomatognathic system is clear in children affected by cleft palate: infants with unrepaired cleft palate under the age of 2 years show a higher incidence of OME [14].

Here, we summarize the current knowledge on this topic, exploring the known relations between ear, TMJ, and dental occlusion, and evidence and hypothesis supporting the efficacy of gnathological treatment in preventing OM in its various forms.

## 2. Aims

The review aims to collect current evidence about the impact of TMJ dysfunction and malocclusion on the onset and course of OM.

The main objectives of the review were to: (I) Describe the embryological, anatomical, and functional relations between the stomatognathic system, the ear, and associated structures; (II) Summarize available evidence of the role of TMJ dysfunction, mandibular malposition. In addition, malocclusion on OM pathogenesis, and the effectiveness of gnathological therapies on preventing OM and ameliorating middle ear function and ventilation; (III) Discuss the actual knowledge on OM pathogenesis, risk factors, and associated comorbidities in light of the possible association with TMJ dysfunction and malocclusion.

## 3. Methods

A narrative review was carried out. The searches were conducted using the PubMed database, using various combinations of the following keywords: (otitis OR otitis media OR acute otitis media OR recurrent acute otitis media OR otitis media with effusion OR Eustachian tube OR Eustachian tube function OR Eustachian tube dysfunction) AND (temporomandibular joint OR temporomandibular joint dysfunction OR temporomandibular disorder OR malocclusion). The search was conducted in October 2022.

To be included in the review, articles had to meet the following criteria: (1) published in peer-reviewed journals; (2) written in English; (3) consider the role of TMJ function or malocclusion or orthognathic treatments in the clinical course of OM in its various form or Eustachian tube dysfunction; (4) only human subjects included.

Titles and abstracts were viewed: the appropriate articles were reviewed, and the inappropriate ones were discarded. References in the selected articles were reviewed and included if not already listed. Further relevant papers were added from the authors’ libraries if not already included. The studies were then grouped according to their content and relevant findings described in narrative chapters.

Details on study selection are summarized in Figure 1.

## 4. Relevant Sections

### 4.1. Shared Phylogenesis and Embryogenesis of Middle Ear and TMJ

The actual anatomy of the middle ear and TMJ structures derives from changes in the splanchno-cranium from reptile to mammal evolution. In reptiles, the mandible is composed of multiple bones, and there is a single bone in the ear. In the mammalian jaw, only the dentary bone was left. The bones of the old joint detached from the mandible and are part of the middle ear bone chain: the articular bone (mandible) became the malleus, and the quadrate bone (cranium) became the incus [15].

In mammals, the maxillary and mandibular bones, the ET, the tympanic membrane, and most middle ear structures develop from the 1st pharyngeal arch [16]. Pharyngeal arches are the segmentations of the anterior part of the developing embryo. The most rostral is the 1st pharyngeal arch, which divides into the maxilla and mandibular arch and will form the upper and lower jaws [16]. The arches are lined externally by ectoderm and internally by endoderm. Each arch is separated by pockets of ectodermal clefts and endodermal pouches. The first pharyngeal endodermal pouch extends to the ectoderm forming the tubotympanic recess that later becomes the ET and the lining of part of the middle ear cavity [16]. The 1st arch is filled with neural crest cells surrounding a mesodermal core: the former originates the skeletal elements of the jaw, connective tissue, tendons, pericytes, and smooth muscle cells of the arch arteries, while the latter gives rise to the arch arteries and most skeletal muscles [17,18,19]. So, the mandible arises from the ventral part of the Meckel’s cartilage, which derives from the 1st arch, while the malleus and incus are from its dorsal part [13]. Proof of this shared origin is the petrotympanic fissure, a canal that connects the middle ear and the TMJ via neurological, vascular, and ligamental structures, which remains after the Meckel’s cartilage is reabsorbed [13]. Meckel’s cartilage plays a role in organizing and forming jointly-located anatomical structures, which, regrettably, are assessed by separate health disciplines [20]. Origin from the 1st pharyngeal arch is also shared by muscles, directly and indirectly, involved in ET function, namely the tensor tympani muscle, the tensor veli palatini (TVP) muscle, the masticatory muscles, the anterior belly of the digastric muscle, the mylohyoid muscle, and the related motor innervation provided by the maxillary and mandibular branch of the trigeminal nerve [21].

Finally, an indirect proof of this shared and interdependent embryologic development is the associated ear anomalies and mandibular dysplasia in the case of congenital malformations such as in congenital microtia, Treacher Collins syndrome, Turner syndrome, or 21 Trisomy. Such conditions also increases the risk of AOM, rAOM, OME, and CSOM [5,22,23,24].

### 4.2. Anatomical Connections between Middle Ear and TMJ and Functional Anatomy

The TMJ is a diarthrosis formed by the union of the temporal bone cavity with the mandibular condyle. The articulation comprises a synovial cavity filled with synovial fluid, articular cartilage, and a capsule covering the same joint. A complex of ligaments and muscular tendons stabilizes and consents mandibular movement, forming a morpho-functional unit. The most relevant are the sphenomandibular ligament, the stylomandibular ligament, the pterygomandibular ligament, the discomalleolar ligament, and the collateral ligament. Among muscles, those which close the jaw are the masseter, temporal, and lateral pterygoid; those which open the jaw are the medial pterygoid, geniohyoideus, mylohyoideus, and digastric [25].

The middle ear, the TMJ, and the mandible are anatomically related by their proximity and direct and indirect connections.

The TMJ and the ear are anatomically close. The cranial surface of TMJ consists of the squamous area of the temporal bone; it is called the glenoid fossa and welcomes the condyle of the jaw. The posterior area of the fossa contains a bone portion called the postglenoid process, which contributes to forming the upper wall of the external acoustic meatus [26]. The proximity of TMJ and the middle ear is also testified by the risk of septic TMJ arthritis complicating an OMA [27]. Communication between the outer ear canal and TMJ, the so-called Huschke foramen, is notably present in 4.6–20% of the population, according to studies [28].

The most direct communication between the middle ear and the TMJ is the abovementioned petrotympanic fissure, which contains the anterior tympanic and deep auricular arteries, the chorda tympani and auriculotemporal nerve, and the disco-malleolar and anterior malleolar ligaments [13].

However, the most relevant aspect in understanding OM pathogenesis is represented by the anatomical and functional relations between ET, TMJ, and mandible. The ET lies immediately below the skull base, runs anteriorly, medially, and inferiorly from the middle ear to the nasopharynx, and is formed of osseous (lateral one-third) and cartilaginous (medial two-thirds) sections [29]. It lies at 34–36 degrees relative to the axial plane and 42 degrees to the sagittal plane in adults. The angle is shallower, and the length is shorter in children than in adults (38 vs. 44 mm) [29].

The ET is collapsed at rest; the oval-shaped lumen opens during swallowing as the result of contraction of the para-tubal muscles on the cartilaginous portion of the Eustachian tube. The TVP muscle is the most important muscle in opening the ET [6,29,30]. Other para-tubal muscles are the levator veli palatini, the tensor tympani, and the salpingopharyngeus, but their role in opening the ET is deemed marginal [29].

The TVP consists of a lateral layer originating from the skull base and a medial layer arising from the lateral lamina of the tubal cartilage. The lateral layer pulls from the skull base to a small tendon going around the pterygoid hamulus and spreading into the aponeurosis of the soft palate. Between the tendon and the hamulus, there is a small bursa. The medial layer of the tensor is situated between the lateral lamina of the tubal cartilage and the medial lamina of the pterygoid process. There are two fasciae that cover the medial and the lateral surface of the TVP muscle. Laterally, there is the Weber-Liel fascia, which separates the TVP muscle from the medial pterygoid muscle. Medially, a fascia runs from the lateral lamina of the tubal cartilage along the lateral surface of the so-called lateral Ostmann fat pad to the salpingopharyngeal fascia, which is also called the von Tröltsch fascia [30].

The function of the TVP muscle is complex for two reasons: (I) its contraction is completely isometric, meaning that the opening function depends on a system of levers that Leuwer et al. call hypomochlia; (II) both TVP layers have dissimilar effects on ET function: whereas contraction of the medial layer opens the eustachian tube by lateralization of the lateral lamina of the cartilage, the lateral layer compresses the lower portion of the tube. Thus, the medial layer supports ventilation, and the lateral layer supports drainage and protection [30]. There are three hypomochlia influencing the TVP muscle: (I) the pterygoid hamulus; (II) the lateral Ostmann fat pad (a thin layer of fat tissue between the ET and the skull base); (III) the medial pterygoid muscle [30].

Finally, two masticatory muscles need to be cited to fully understand the relations between ET, TMJ, and the mandible: the medial and lateral pterygoid muscles. The medial pterygoid muscle is a chewing muscle that closes the mouth and helps protrude the mandible. Its contraction causes a posteromedial movement of the tensor toward the cartilage, increasing the tubal opening pressure. Inversely, the opening of the ET is facilitated by the relaxation of the medial pterygoid due to an anterolateral movement of the TVP while opening the mouth. Due to their connection via the Weber-Liel fascia, the TVP and medial pterygoid muscles form a mechanical functional unit. Simultaneous relaxation of the medial pterygoid muscle, as well as contraction of the TVP muscle by yawning, can be used as a maneuver for the physiologic active tubal opening during external pressure changes, such as the landing of an airplane. Morphological or functional alterations of the medial pterygoid muscle, for example, in craniomandibular disorders, may change the muscular compliance of the eustachian tube [30].

The lateral pterygoid muscle is a single unit divided into two bellies: the superior belly inserts anteriorly on the infratemporal surface of the sphenoid bone’s greater wing and the pterygoid process lateral wing, and posteriorly on the TMJ disc and mandibular condyle; the inferior belly anteriorly attaches on the lateral area of the pterygoid process lateral wing, lateral area of the pyramidal process, and part of the maxillary bone tuberosity, and posteriorly to the mandibular condyle and TMJ capsule [31].

In normal subjects, cine CT imaging acquired during a sequence of swallows demonstrated a discrete air bolus progressing superiorly toward the middle ear in a peristaltic-like movement with medial and lateral pterygoid muscle contractions [32]. The lateral pterygoid muscle movement occurred during volitional jaw movement and normal swallows. While lateral pterygoid contraction would be necessary for a subject to protract the mandible, such movement is not intuitively necessary for swallowing. The lateral pterygoid appears to border and enhance the dilation of the tympanic ET, though it is not as closely associated with the ET as the medial pterygoid. So, contraction of the medial and lateral pterygoids may alter the convexity of the anterolateral ET, thus allowing TVP contraction to have a greater effect.

### 4.3. Role of the Stomatognathic System in the Pathogenesis and Treatment of the OM

The anatomical and physiological relations between middle ear structures, particularly ET, TMJ, and the mandible, and their interdependent development corroborate the role of the gnathic system in the pathogenesis of OM.

The shortening and thickening of the mastication medial pterygoid muscle in over-closed jaw positions exert a lateral pulling force on the TVP muscle and the ET. An increase in medial pterygoid transverse volume can result from an open-closed position depending on muscle length and tension variation. Thus, the medial pterygoid muscle activity could change muscle tension direction together with another two structures contacting the TVP (pterygoid hamulus and Ostmann’s fatty tissue), which may influence middle-ear ventilation by Eustachian tube compliance [13]. The muscular mass compression and lateral bunching of the TVP and associated structures could result from a deep overbite, medial pterygoid shortening, hypertonicity, or spasm [13]. The hypertonicity of masticatory muscles due to a TMD could make the TVP and tensor tympani muscles hypertonic through the shared innervation by the mandibular branch of the trigeminal nerve [13].

McDonald et al. conducted a radiological study on 16 adults with or without ET dysfunction who have undergone a cine computed tomography during swallowing or other jaw movements. Images obtained from subjects with ET dysfunction displayed a smaller or absent air bolus passing through the ET and a little, if any, observable medial pterygoid movement. They conclude on the amplificatory role of the masticatory muscles on ET opening and closing [32].

As electromyographic studies suggest, the activity of the masticatory muscles could be altered by a malocclusion. The involved side and muscular group depend on the type of malocclusion [33]. A significant reduction in maximal mouth opening performance (strength and endurance) is observed in patients with TMD [34]. These elements appear relevant in the pathogenesis of OM since the opening of the ET is facilitated by the relaxation of the medial pterygoid due to an anterolateral movement of the TVP while opening the mouth.

These considerations could help evaluate eligibility for ET dilation with catheter ballooning, a procedure that has recently emerged to treat ET dysfunction. This procedure has shown good efficacy in updated metanalyses; however, 30 to 50% of subjects seem non-responding to this therapy at a medium-term follow-up [35]. The motivation could rely on pterygoid muscle dysfunction on a malocclusion basis: even if the procedure is immediately effective, the undergoing abnormal activity of crewing muscles could impair the operation results in the long term. Clinicians proposing ET dilation should probably be aware of these possible pathogenetic mechanisms that could reduce the success rate. A thorough evaluation of the gnathic function could bring a tailored indication of ET ballooning.

Several studies on the pediatric population support the relationship between stomatognathic structures and function and the risk of OM. In a study on 105 infants, McDonnell et al. found deep bite an independent risk factor for ET dysfunction, with a 2.8 times higher odds ratio compared to children with normal occlusion [36]. In a similar study, the odds ratio of developing an ET dysfunction reached 10.6 in children with a deep bite [13]. Costen originally postulated that the overclosure that occurs in patients with a reduced vertical dimension of occlusion (deep bite) causes bunching of the TVP muscle, which is the only muscle that acts directly on the ET [37]. The deep bite has been deemed a predisposing factor to OM also due to posterior condyle displacement and subsequent inflammation and muscle spasm in the retro-discal tissue and muscles surrounding the TMJ. These alterations could also block lymphatic channels from the ear to the mandibular condyle via the petro-tympanic fissure [38]. In addition, the nasopharyngeal dimensions of children with rAOM are reported to be smaller [39]. Children with Class II malocclusions have been found to have smaller nasopharyngeal dimensions than those with other occlusal Classes. Children with a Class II dental and skeletal relationship usually have a large overjet and lower incisors that tend to erupt until they contact the palatal mucosa, thereby creating a deep bite (Figure 2) [40].

Greater alterations in the craniofacial morphology and in the mandibular-maxillary relation were found in children with rAOM younger than 4 years compared to healthy control in a study by Gremba et al. [41]. Specifically, in the rAOM group, the palate and pterygomaxillary fissure are inferiorly displaced, the middle cranial base and external acoustic meatus are posteriorly displaced, and the posterior cranial base is shortened in comparison to the Control group. The picture is coherent with the changes observed in the deep bite. The association disappears in toddlers older than 6 years: this is consistent with the clinical observation that excess rAOM risk often resolves around 6 years of age and the hypothesis that this resolution is due, in part, to age-related craniofacial changes. Di Francesco et al. found similar results in a study on 67 children from 5 to 10 years old whose craniometric findings were compared according to the presence of OME [42]. Subjects with OME presented differences in the morphology of the face regarding these measures: anterior cranial base length, upper facial height, size of the hard palate, facial depth, facial axis, mandibular length, and inferior pharyngeal airway. In particular, facial depth and facial axis were found to be greater in the OME group than in the control group, similar to what happens in the case of a deep bite. The mandibular body length was shorter in the OME group, as in retruded mandibles [43]. The high prevalence of malocclusion and associated TMD in children is consistent with the epidemiology of OM in the early ages. Open bite, deep bite, and posterior crossbite seemed to be the most important associated with TMD [44].

If alterations in the stomatognathic system could increase the risk of OM, it is also possible that treating the predisposing conditions benefits ET function, preventing middle ear inflammations.

In 1988 Marasa and Ham presented four cases of children affected by retrognathic bite and OME or rAOM, who resolved their otologic complaints after receiving orthodontic treatment [45]. Similar reports come from Branam and Mourino [46]. Figure 2 depicts an analogous case.

A study led by the first author (E.B.) assessed the effect of orthognathic treatment on children with rAOM [8]. The clinical outcome (number of acute recurrences in 12 months) of 61 consecutive children treated medically for rAOM was analyzed. Children underwent an odontostomatologic evaluation, a fiberoptic endoscopy, and skin-prick tests. Of these, 32 children were diagnosed with dental malocclusion and treated with a mandibular repositioning plate (Figure 2). Dental malocclusion was ruled out in the other 29 patients with rAOM, and they were used as controls. The two groups were homogeneous in terms of sex, exposure to rAOM risk factors, skin test results, and adenoid hypertrophy, while age was significantly higher in the treatment group. Age, sex, exposure to RAOM risk factors, adenoid hypertrophy, and skin test results were not associated with rAOM outcome, while those treated for dental malocclusion showed a significantly lower number of acute recurrences at both univariate and multivariate analysis. The main weakness of the study concerned the control group, which included children with rAOM and without dental malocclusion, while the ideal control cohort would have included children with rAOM and dental malocclusion not wearing the oral device. This exact match could not be achieved because of ethical concerns; however, the comparability of the two groups for demographic and OM-related risk factors strengthens the results.

The beneficial effects of gnathological treatments on middle ear conditions are also stated in works considering maxillary expansion (Figure 3).

Kilic et al. report significant hearing improvement and air-bone gap reduction after semi-rapid maxillary expansion in a group of 19 adolescents, and these changes remained relatively stable during the last two periods. Middle ear volume increased statistically significantly after maxillary expansion and continued to increase until the end of treatment. After maxillary expansion, the TVP muscle is stretched, and its contractible activity improves with consequent more effective vectorial forces [47]. Singh et al. evaluated the otological effects of maxillary expansion in children with non-cleft and bilateral cleft lip palate: treatment yielded a significant increase in the hearing levels and middle ear volumes of all non-cleft and bilateral cleft lip palate patients with normal hearing levels and with mild conductive hearing loss, while only non-cleft children benefitted from the therapy among subjects with moderate hearing loss [48]. Results are confirmed in a systematic review on this topic by Fernandes Fagundes et al. [49].

## 5. Discussion

Otitis media in its various forms is a relevant clinical problem. Involved pathogens are well renowned, namely Streptococcus pneumoniae, Haemophilus influenzae, Streptococcus pyogenes, Moraxella catarrhalis, and Staphylococcus aureus. The introduction of vaccination against S. pneumoniae has been associated with a reduction in OM caused by these bacteria and a rise of the other pathogens as causative agents [5]. The appropriateness of antibiotic therapy in childhood is debated, and AOM and rAOM constitute one of the most frequent indications for antibiotic prescription, reaching 25% of overall causes in some studies [50]. Recommendations from Italian guidelines on rAOM are to immediately administer antibiotics in case of recurrence [51]. Similarly, recurrence is an indication of antibiotic therapy in American guidelines [7]. Evidently, the therapeutic recommendation is directed to resolve the acute episode, which is usually easy to treat but does not address the underlying causal or predisposing factors truly responsible for the recurrences, which represent the core clinical challenge.

### 5.1. Prevention

From a preventive perspective, risk factors for AOM are classified as modifiable and unmodifiable. Vaccinations, xylitol, probiotics, and vitamin D supplementation show low evidence of efficacy [7,52,53]. In the case of rAOM, Italian and American guidelines advise against antibiotic prophylaxis and propose trans-tympanic tubes as the last option for otherwise intractable forms [7,51]. A randomized clinical trial evaluating prevention strategies for rAOM showed that trans-tympanic tubes have similar results compared to a placebo, and they simply drain the effusion fulfilling the function of the ET. Trans-tympanic tubes are also associated with permanent complications, such as perforations, tympanosclerosis, and tympanic atrophy [54].

In the pathogenetic ET-TMJ model discussed above, if a single episode of AOM could be attributed to a sporadic viral or bacterial colonization of the middle ear coming from the upper airway tract, rAOM or OME could be interpreted as a recurrent infection or persisting effusion due to relapsing or chronic TVP dysfunction, mediated by a dysfunctional bite [55]. The middle-ear transudation effect is due to hypoventilation and abnormal gas exchange when the ET becomes blocked and impeded in regulating pressure. The ET normally maintains a closed position at rest, protecting the middle ear from retrograde nasopharynx microflora flow during rapid fluctuations in nasopharyngeal pressure associated with breathing, swallowing, coughing, sneezing, and nose-blowing. Children’s Eustachian tube dysfunction plays an important role because of the anatomical configuration of the Eustachian tube (short, horizontal, and wide lumen) or the TVP open-closed dynamics when spastic or trapped by the neighboring pterygoid muscle [45]. So, treating a predisposing stomatognathic dysfunction could ameliorate ET function and prevent recurrent episodes of AOM [8].

From this perspective, an orthognathic treatment could be seen as a form of primary, secondary, or tertiary prevention, depending on the time point it is administered in the natural history of OM: as primary prevention when malocclusion is intercepted before an episode of AOM take place; as secondary prevention to prevent relapses, chronicization, CSOM, and complications such as cholesteatoma or TMJ septic arthritis; as tertiary prevention to maintain the results of otological surgery when ventilation routes are restored or in case of radical procedures such as for extensive cholesteatomas [27,56,57].

This conceptual framework shed new light on renowned modifiable and unmodifiable risk factors with some changes to their classification.

### 5.2. Unmodifiable Risk Factors

Age, male sex, Caucasian ethnicity, genetic factors, school-attending siblings, prematurity, immunological deficiency, atopy, and anatomical factors such as ET dysfunction and craniofacial anomalies are included among unmodifiable risk factors for AOM [58].

As explained above, ET function can be influenced by the mandibular position and dental occlusion; therefore, it should be listed among the modifiable risk factors. The characteristic anatomy of the ET in childhood is thought to be the main reason for its dysfunction since it is shorter and with a shallower angle compared to adults [3]. Nevertheless, this does not justify the frequent unilateral forms. Moreover, the ET angle does not differ between children with and without otitis media [3]. Adenoidectomy is largely performed to reduce ET blockage and ameliorate middle ear ventilation, but its actual effectiveness in preventing rAOM is controversial [7]. Contrariwise, a malocclusion with a retruded or deviated bite and TMJ dysfunction could justify a bilateral or unilateral alteration of TVP contractility and consequent rAOM or OME, which could be bilateral in case of bilateral retrusion, unilateral in case of lateral bite deviation [8]. Myo-functional treatments such as ET Rehabilitation Therapy have been proposed, and they have shown favorable outcomes in children affected by OME, oral breathing, and swallowing disorders, with measurable effects on TVP activity [59,60]. Mandibular repositioning with an oral appliance and subsequent orthodontic treatment could be seen as a long-term myo-functional treatment since they permanently change the vectorial forces of the medial pterygoid muscles and TMG ligaments, with possibly more durable effects compared to ET Rehabilitation Therapy, whose ending could be followed by a relapse once the muscular dysfunction re-establishes due to malocclusion.

As discussed in the “Relevant Sections”, craniofacial anomalies are frequently associated with rAOM and OME. Fortunately, cases of craniofacial malformations are rare. Contrariwise, abnormal facial shape and position of facial components are frequently seen in malocclusion, which can be adjusted with an orthognathic treatment. Therefore, most cases addressed as “unmodifiable” could be easily treated with non-invasive interventions [8].

Genetic factors are implied in a familiar history of rAOM or OME, especially for homozygotic twins [58,61]. Moreover, the craniomandibular shape and malocclusion could be inherited [62]. Occlusal relations and mandibular position can be changed by gnathological remediation [63]. Correcting the morphologic predisposing factor could reduce rAOM and OME. Therefore, a variable number of genetic factors could be re-classified as “modifiable” risk factors (Figure 4).

### 5.3. Modifiable Risk Factors

Obstructive sleep apnoea syndrome (OSAS), gastro-oesophageal reflux disease (GERD), school attendance, and exposure to smoke and air pollution are among the modifiable risk factors.

Exogenous factors such as smoke and air pollution need an entryway to the nasopharynx, ET, and middle ear. Malocclusion may predispose to oral breathing due to difficulty in keeping the mouth closed and muscular fatigue [64]. In the case of oral breathing, a large amount of breathed air bypasses the nasal filter and reaches the pharyngeal mucosa, potentially favoring inflammation, infections, and adeno-tonsillar reactive hypertrophy with subsequent OSAS. In fact, mouth-breathing children show higher levels of oxidative-stress-related salivary proteins (lactoylglutathione lyase and peroxiredoxin-5) and opportunistic bacteria with significant differences in nasopharyngeal and oral microbiota, while immune-related proteins (integrin alpha-M and proteasome subunit alpha type-1) are down-regulated [65]. Chronic inflammatory stimuli in the upper aerodigestive tracts impair the immunological patency of the Waldeyer’s ring, causing ineffective and pro-inflammatory responses to pathogens [66]. This is consistent with the use of probiotics as a prevention line [7]. Considering this, oral breathing due to malocclusion could start a vicious cycle that leads to adeno-tonsillar hypertrophy and OSAS, with the worsening of nasal breathing and a higher risk of rAOM. Addressing the malocclusion could correct primary oral breathing, permitting the nasal filter to execute its physical and immunological functions, thus preventing pharyngeal inflammations, dysbiosis, adeno-tonsillar hypertrophy, and finally, rAOM and OME.

Speaking of OSAS, it is thought to increase the risk of rAOM and OME. However, the effectiveness of OSAS treatment on OM outcomes is still debated. Robison et al. showed an increased incidence of ET dysfunction in children with OSAS and a decreased need for myringotomy and tympanic tube placement after surgical interventions for treating OSAS [67]. Ungar et al. found similar results concerning the prevalence of ET dysfunction in the OSAS population; however, expansion sphincter pharyngoplasty did not change ET function in the long term [68]. An enlightening hint comes from the work of Di Francesco et al.: they evaluated craniofacial morphology and the presence of OME in patients with chronic upper airway obstruction due to tonsil and adenoid enlargement. Interestingly, children with OME showed significant differences in craniofacial alterations and narrower inferior pharyngeal airways [42]. This is consistent with the well-renowned role of malocclusion and mandibular malposition in the pathogenesis and treatment of OSAS: the same predisposing anatomical features could lead to sleep breathing disorders on one side, and ET dysfunction with rAOM or OME on the other, and treatment with oral appliances could prevent both conditions. 68 Noticeably, the oral devices used in the study of Villa, Bernkopf, et al. (2002) on malocclusion and OSAS and Bernkopf et al. (2016) on malocclusion and rAOM are the same [8,69].

Finally, GERD is listed among the potential risk factors for OM due to the inflammatory effects on ET. The prevalence of GERD in children with OME and rAOM may be higher than the overall prevalence for children. The presence of pepsin/pepsinogen in the middle ear could be related to physiologic reflux. Nevertheless, a cause-effect relationship between pepsin/pepsinogen in the middle ear and OM is unclear, and anti-reflux therapy for OM cannot be endorsed based on existing research [70]. Obstructive sleep apnoea syndrome and GERD have also been linked. Interestingly, in a study on children with OSAS and GERD, Cohen and Tirosh observed that when apnoea and reflux were timely related, apnoea preceded reflux in 93.6% of the episodes and it was obstructive in 66.8% of episodes and mixed in 33.2%, while central apnoea was not recorded preceding reflux [71]. In the framework depicted in this article, a malocclusion and a mandibular malposition could be postulated to cause both OM and episodes of obstructive apnoea, with the former generating a negative pressure on the gastroesophageal tract due to communicating vessels, thus inducing GERD.

### 5.4. Future Directions

Randomized controlled trials are advocated to determine the quantitative contribution of malocclusion in the onset of OM and the effectiveness of orthognathic treatment in preventing relapses and chronicization. Groups that are homogeneous for confounding factors and stratified for ages would be recommended.

Current research is not evaluable with metanalysis due to variable endpoints, non-standardized treatments, and lack of uniformity in defining fundamental features such as ET dysfunction. When solid evidence is collected, a clear profile of risk should be included in guidelines to provide physicians with practical tools to screen for possible malocclusion predisposing to or sustaining OM. This would aim to recognize the orthodontist’s role and include the gnathological evaluation and treatment in the integrated management of OM.

### 5.5. Highlights

The sequent key points emerge from our review:the external and middle ear, the TMJ, the mandible, the ET, and associated musculature and ligaments share interdependent embryogenesis and development;these structures have multiple anatomical connections. In particular, ET, TMJ, and masticatory muscles are strictly connected;ET function is influenced by the mandibular position and masticatory muscles function; the role of the medial pterygoid muscle is particularly emphasized;mandibular position and malocclusion are related to a higher risk of AOM and rAOM; this is particularly evident in children with a deep bite and a crossbite;the studies on the effectiveness of orthognathic treatments in preventing rAOM show encouraging results;TMJ dysfunction and malocclusion are strongly related to known predisposing factors for AOM, rAOM, and OME; addressing them could further protect the patient from ear inflammations.

## 6. Conclusions

The current literature supports malocclusion with mandibular malposition and TMJ dysfunction as a predisposing factor to ET dysfunction due to alterations of TVP activity. This persisting condition could lead to rAOM, OME, and potential chronicization of OM. Gnathological, conservative treatments with removable oral appliances improve TVP and ET function, consenting middle ear ventilation and preventing relapsing or persistent forms of OM. Otolaryngologists, pediatricians, and physicians should evaluate the occlusal status of patients, especially when recurrent inflammations are present. Otitis media should be regarded as a multidisciplinary pathology requiring multidisciplinary management.

## Figures and Tables

**Figure 1 medicina-58-01806-f001:**
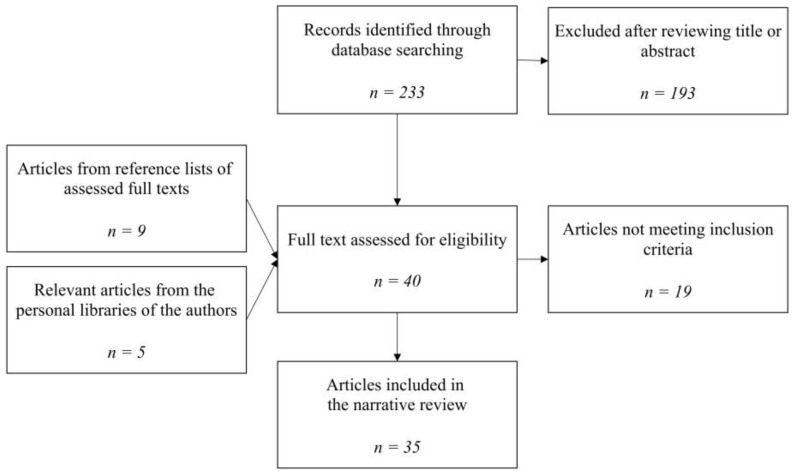
Study selection.

**Figure 2 medicina-58-01806-f002:**
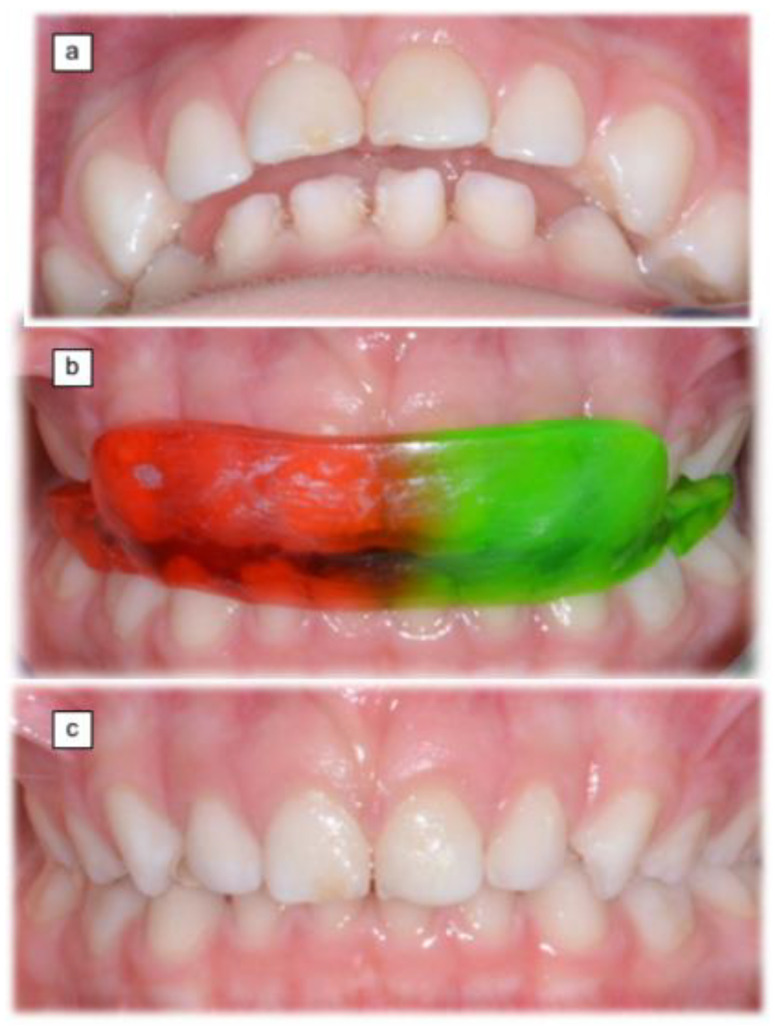
A 4-year child affected by rAOM and severe OSAHS (Apnea-Hypopnea Index = 10.8). (**a**) Dental malocclusion with severe overjet. (**b**) Mandibular repositioning with the oral appliance: the resolution of rAOM episodes. (**c**) Six-month result: no AOM recurrence, Apnea-Hypopnea Index = 0.4.

**Figure 3 medicina-58-01806-f003:**
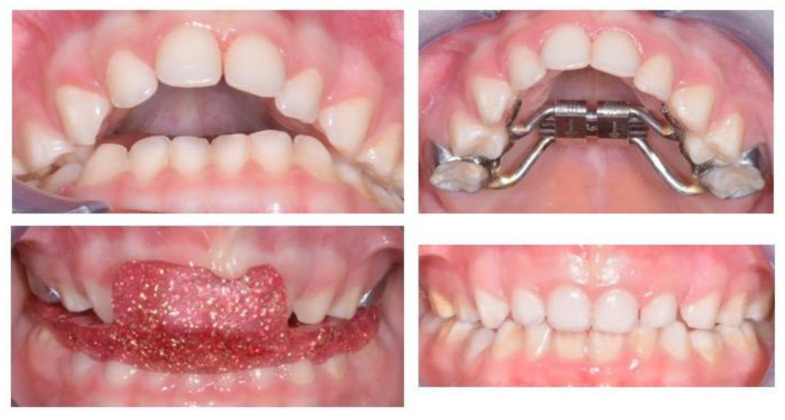
A child affected by rAOM, recurrent impacted cerumen, and OSAHS. A severe retruded bite, overjet, and ogival palate are present. The oral appliance is adaptable to coexist with a maxillary expansion device.

**Figure 4 medicina-58-01806-f004:**
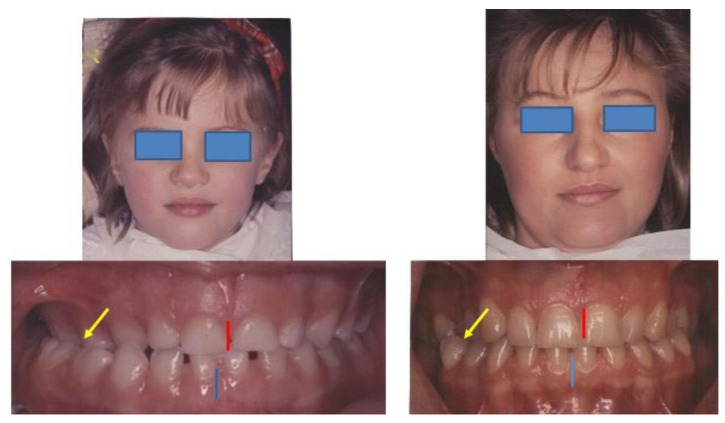
Dental malocclusion with lateral mandibular deviation and right cross-bite. A child with rAOM and her mother with a previous history of rAOM and current periodic dizziness. The right ear was affected in both cases.

## Data Availability

Not applicable.

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
