# Peer review of "Temporomandibular Joint and Otitis Media: A Narrative Review of Implications in Etiopathogenesis and Treatment"

_medicina, 2022, doi:10.3390/medicina58121806_

Round 1

Reviewer 1 Report

Brief summary:
The current paper is a review of the relationship between the anatomical structures surrounding the TMJ as they relate to occlusion. The paper was well written. However, there are some points of clarification that need to be addressed.

Specific comments:

1. The authors may need to clarify if this is a narrative or a scoping or a systematic review. This should reflect in the title. (Lines 1-3)

2. Despite being a narrative review, the authors may still clarify and include the inclusion criteria. A reporting guideline may also be used.

3. The Italian guidelines on rAOM on the immediacy of antibiotics was mentioned. What does the international guideline or consensus entail? (Lines 296-297)

Author Response

Dear Editors,

Dear Reviewers,

We thank you for reviewing our work.

We believe that the raised issues have further improved the quality of it, and we sincerely thank you for these contributions.

All the suggested corrections have been incorporated into the manuscript.

All the changes have been highlighted using the “Track changes” function.

Herein is a detailed point-by-point response letter to address each issue raised by the reviewer.

REVIEWER 1

  1. The authors may need to clarify if this is a narrative or a scoping or a systematic review. This should reflect in the title. (Lines 1-3)

The title has been modified and the type of review specified.

  1. Despite being a narrative review, the authors may still clarify and include the inclusion criteria. A reporting guideline may also be used.

Thank you for this observation. An “Aims” and a “Methods” sections have been added that address this issue.

  1. The Italian guidelines on rAOM on the immediacy of antibiotics was mentioned. What does the international guideline or consensus entail? (Lines 296-297)

To our knowledge, there is not an international consensus on the management of rAOM, but only on otitis media with effusion. However, a reference to American guidelines has been added to confirm the general approach, which considers recurrence as an indication for antibiotic therapy.

Reviewer 2 Report

As the authors of this study mentioned, the effects of TMJ and dental malocclusion in otitis media are somewhat overlooked. In fact, ENT specialists are not very familar with the abnormality of stomatognathic system. This review is positive as a good opportunity to examine the impact of TMJ in otitis media.

Although it is a narrative review, it is too narrative to catch the implication. Prior to the conclusion section, highlighting the main contents will help readers understand the clinical implications of this study.

Recently, E-tube cathether ballooning is performed in adult patients when eardrum retraction is repeated due to the obstruction of E-tube. Is there anything we need to consider when performing this procedure? If the medial pterygoid muscle activity has an impact, in a person with the dysfunction of the muscle activity, I think there may be no significant improvement even though the ballooning is performed. Maybe this is the reason that E-tube ballooning is not successful in some patients.

Line 292-293: Traditionally, S pneumoniae, H influenzae, and M catarrhalis have been knonwn as causative bacteria, but there are likely to be some changes. In our country, owing to pneumococcal vaccination in children, the rate of detection of S pneumoniae as a causative agent has been dramatically reduced.

Otherwise, those are trivial things

Line 31: “The etiopathogenesis is still debated although is thought to be multifactorial” → “The etiopathogenesis is still debated, though it is thought to be multifactorial”

Line 91, line 328: does it mean AOM? or other? Please write full name at first.

Line 142-143: “eustachian” → “Eustachian”

Line 191: Since TMD appeared first, please write full name here. It probably means temporomandibular disorder.

Line 263: “RAOM” → “rAOM”, maybe

Line 293: “e” → “and”

Line 431: “metanalysis” → “meta-analysis” 

Author Response

Dear Editors,

Dear Reviewers,

We thank you for reviewing our work.

We believe that the raised issues have further improved the quality of it, and we sincerely thank you for these contributions.

All the suggested corrections have been incorporated into the manuscript.

All the changes have been highlighted using the “Track changes” function.

Herein is a detailed point-by-point response letter to address each issue raised by the reviewer.

REVIEWER 2

As the authors of this study mentioned, the effects of TMJ and dental malocclusion in otitis media are somewhat overlooked. In fact, ENT specialists are not very familiar with the abnormality of stomatognathic system. This review is positive as a good opportunity to examine the impact of TMJ in otitis media.

Although it is a narrative review, it is too narrative to catch the implication. Prior to the conclusion section, highlighting the main contents will help readers understand the clinical implications of this study.

Thank you for this comment. An “highlights” section has been added.

Recently, E-tube cathether ballooning is performed in adult patients when eardrum retraction is repeated due to the obstruction of E-tube. Is there anything we need to consider when performing this procedure? If the medial pterygoid muscle activity has an impact, in a person with the dysfunction of the muscle activity, I think there may be no significant improvement even though the ballooning is performed. Maybe this is the reason that E-tube ballooning is not successful in some patients.

This is an interesting observation. Some consideration on this aspect has been added in the main text.

Line 292-293: Traditionally, S pneumoniae, H influenzae, and M catarrhalis have been knonwn as causative bacteria, but there are likely to be some changes. In our country, owing to pneumococcal vaccination in children, the rate of detection of S pneumoniae as a causative agent has been dramatically reduced.

Thank you for this observation. A note about this matter has been added.

Otherwise, those are trivial things

Line 31: “The etiopathogenesis is still debated although is thought to be multifactorial” → “The etiopathogenesis is still debated, though it is thought to be multifactorial”

Line 91, line 328: does it mean AOM? or other? Please write full name at first.

Line 142-143: “eustachian” → “Eustachian”

Line 191: Since TMD appeared first, please write full name here. It probably means temporomandibular disorder.

Line 263: “RAOM” → “rAOM”, maybe

Line 293: “e” → “and”

Line 431: “metanalysis” → “meta-analysis”

These errors have been fixed.
